# Edible Fruits from the Ecuadorian Amazon: Ethnobotany, Physicochemical Characteristics, and Bioactive Components

**DOI:** 10.3390/plants12203635

**Published:** 2023-10-21

**Authors:** Maritza Sánchez-Capa, Mireia Corell González, Carlos Mestanza-Ramón

**Affiliations:** 1Departamento de Agronomía, Universidad de Sevilla, ETSIA Crta. de Utrera Km 1, 41013 Seville, Spain; mcorell@us.es; 2Research Group YASUNI-SDC, Escuela Superior Politécnica de Chimborazo, Sede Orellana, El Coca 220001, Ecuador; 3CSIC Associate Unit, “Uso Sostenible del Suelo & Agua en Agricultura”, Universidad de Sevilla IRNAS, 41013 Seville, Spain

**Keywords:** Ecuador, biodiversity, tropical products, underutilized crops

## Abstract

In the Ecuadorian Amazon region, there are various types of edible fruits that have distinct qualities and benefits. Understanding the uses, properties, and functions of these fruits is important for researching products that are only available in local markets. This review aims to gather and summarize the existing scientific literature on the ethnobotany, physicochemical composition, and bioactive compounds of these native fruits to highlight the potential of the region’s underutilized biodiversity. A systematic review was carried out following the PRISMA methodology, utilizing databases such as Web of Science, Scopus, Pubmed, Redalyc, and SciELO up to August 2023. The research identified 55 edible fruits from the Ecuadorian Amazon and reported their ethnobotanical information. The most common uses were fresh fruit consumption, preparation of typical food, and medicine. Additionally, nine native edible fruits were described for their physicochemical characteristics and bioactive components: *Aphandra natalia* (Balslev and Henderson) Barfod; *Eugenia stipitate* McVaugh; *Gustavia macarenensis* Philipson; *Mauritia flexuosa* L.f; *Myrciaria dubia* (Kunth) McVaugh; *Oenocarpus bataua* Mart; *Plukenetia volubilis* L.; *Pouteria caimito* (Ruiz and Pav.) Radlk.; and *Solanum quitoense* Lam. The analyzed Amazonian fruits contained bioactive compounds such as total polyphenols, flavonoids, carotenoids, and anthocyanins. This information highlights their potential as functional foods and the need for further research on underutilized crops.

## 1. Introduction

The Amazon is one of the most biodiverse ecosystems in the world; one-tenth of all living species are found in this biome [1]. It covers 44% of the surface of South America and provides invaluable ecosystem and social services such as climate regulation, human culture of indigenous peoples, gene banks, and freshwater reservoirs [2]. It is made up of eight countries: Bolivia; Brazil; Colombia; Ecuador; Guyana; Peru; Suriname; and Venezuela [3,4]. The biodiversity present in the Amazon is a source of plants for human nutrition, including native and introduced fruits domesticated by indigenous populations, which are mostly collected and cultivated for consumption in local markets [5].

Most tropical fruit production comes from Latin American and Caribbean countries. Brazil, Ecuador, México, and Costa Rica are the main producers and exporters of fresh tropical fruits worldwide [6]. However, fruits in the Ecuadorian Amazon are subsistence crops, mainly destined for self-consumption and livestock feed, with low productivity due to poor field management, limited use of technology, and lack of organization of value chains [7]. Fortunately, the consumption of fruits and vegetables is promoted by international and governmental organizations due to their benefits for human health [8]. The need to identify new sources of bioactive compounds has increased, and with it, the research and consumption of native or exotic tropical fruits, which have been shown to have minerals, vitamins, fibers, phenolic compounds, and pigments [9].

Bioactive components of fruits, such as vitamins, carotenoids, and phenolic compounds, are associated with the prevention of chronic pathologies, including cancer, cardiovascular diseases, type 2 diabetes, and Alzheimer’s disease [10]. These components are characterized by their antioxidant properties, anti-inflammatory, antiallergic, antimicrobial, anticancer, antiviral, and antimutagenic activities [11]. Nowadays, profiling the bio-chemical and bioactive components of fruits is essential to identify their health benefits [12]. Unfortunately, Amazonian fruits marketed at local fairs, which are part of the cultural identity of the people [7], lack scientific exploration of their agro-industrial, nutritional, and functional potential [9].

Local markets are a source of information on how botanical resources are used in different regions and on traditional knowledge in general [13]. Among the human groups of the Amazon region, the indigenous people often have greater ethnobotanical knowledge. The main use of fruits in Amazonian communities is for food, and, in some cases, they are considered medicinal [14]. The diversity of species in a region does not mean greater use by the local population; generally, the most abundant species in the ecosystem are used, with a predominance of easily accessible species [15]. Traditional crops and knowledge about their use have been lost with the domestication of species for agriculture. There is a record of 1561 species of edible plants in Ecuador, which corresponds to 9% of the country’s flora [16].

Once the importance of the bioactive profiles of fruits and their nutritional and functional properties have been established, and considering their high diversity in the Ecuadorian Amazon region, there has been little scientific exploration, which has prevented the deepening of new knowledge and their correct use. In this sense, the present work aimed to synthesize the available information on ethnobotany, physicochemical composition, and bioactive compounds of edible fruits present in the Ecuadorian Amazon to know and promote their importance, as well as to evaluate their existing biodiversity.

## 2. Results

After implementing the methodology, a thorough review of 66 documents was conducted, out of which 27 were considered relevant for the selection of edible fruits from the Ecuadorian Amazon and ethnobotanical description [7,17,18,19,20,21,22,23,24,25,26,27,28,29,30,31,32,33,34,35,36,37,38,39,40,41,42] (Appendix A). The remaining 39 documents were utilized to describe the physicochemical features and bioactive components of the nine chosen fruits (Table 1).

The results of this work are presented in three subsections: the first one shows an analysis of the use, origin, and location of the fruits in the Ecuadorian Amazon, responding to the objective of analyzing their ethnobotany in a preliminary way. In the Section 2, a brief description of the most studied Amazonian fruits in Ecuador was made, and their physicochemical characteristics were compiled, identifying their possible management. In the Section 3, the bioactive components analyzed in fruits sampled in the Amazon region of Ecuador were presented to know the potential of these products as functional foods.

### 2.1. Ethnobotany

The analysis of the information made it possible to identify 55 species of edible fruits studied in the Ecuadorian Amazon (Appendix A). The provinces in which the greatest number of studies were identified were Orellana and Sucumbíos (Figure 1). The category of use after food was medicine, followed by animal food and material. The form in which the fruits are usually consumed is fresh fruit, and the part that was reported as the most used was the pulp. Most of the fruits originate from native species (32) rather than being introduced (23). There was a great biodiversity of fruits used; the works analyzed identified 26 families of edible fruits in the Ecuadorian Amazon. The family with the greatest number of species was Arecaceae (10 species), followed by Rubiaceae (5 species), Malvaceae, and Myrtaceae (4 species).

### 2.2. Physicochemical Characterization

In this section, a brief description of the nine edible native fruits of the Ecuadorian Amazon were selected in this review: *Aphandra natalia*; *Eugenia stipitate*; *Gustavia macarenensis*; *Mauritia flexuosa*; *Myrciaria dubia*; *Oenocarpus bataua*; *Plukenetia volubilis*; *Pouteria caimito*; *Solanum quitoense* (Table 2). At the same time, the information corresponding to morphometric characteristics and nutritional composition is summarized.

Based on the physicochemical characteristics listed in Table 3, it is possible to distinguish between two groups of fruits. The first group has oily, compact, fibrous pulp and includes *Aphandra natalia*, *Gustavia macarenensis*, *Mauritia flexuosa*, and *Plukenetia volubilis*. The second group has sweet and sour juicy pulp and includes *Eugenia stipitata*, *Myrciaria dubia*, *Oenocarpus bataua*, *Pouteria caimito*, and *Solanum quitoense*, which have been studied with the interest of relating quality characteristics to agronomic and storage conditions.

*Solanum quitoense*, in addition to the data reported in Table 3, has information on respiration rate (46.1 mgCO_2_/kg⋅h), ethylene production (1.6 mg C_2_H_4_/kg⋅h), and pH (2.96 ± 0.20–3.3); this is probably because it is one of the fruits cultivated in the Ecuadorian Amazon that is marketed nationally, as well as yellow pitahaya, which is grown mainly in the province of Morona Santiago, city Palora, and it is mainly destined for exported [36]. Information on fruit quality characteristics such as pH, soluble solids, titratable acidity, and flavor are subject to variations in environmental conditions such as temperature and rainfall [91]. *Plukenetia volubilis* has high protein content, while *Aphandra natalia*, *Gustavia macarenensis*, and *Plukenetia volubilis* all have lipid content above 30%, indicating their potential as sources of vegetable fats.

### 2.3. Bioactive Compounds and Antioxidant Capacity

Identifying and quantifying bioactive components can greatly impact the recognition and commercial success of fruits. Açaí, for example, was relatively unknown in the 1990s but became the third most popular fruit juice in the U.S. by 2008. Its high antioxidant capacity earned it the label of a superfood [92]. The bioactive compounds reported by the research on Ecuadorian Amazonian fruits are mainly vitamin C, polyphenols, flavonoids, carotenoids, and total anthocyanins (Table 4).

In addition to the bioactive compounds shown in Table 4, *Oenocarpus bataua* reported transresveratrol content (pulp:1.94 ± 0.10, peel:7.98 ± 0.01, seed: 12.33 ± 0.02 mg/g d.w.), and *Gustavia macarenensis* reported as non-detected [27]. This form of resveratrol is biologically active in the human body [95]; it is present in plant foods and recognized for its health benefits such as cardioprotective, antidiabetic, antiobesity, antiaging, and neuroprotective due to its antioxidant, anti-inflammatory, and autophagy-modulating properties [96]. Pulp oil from the fruits *Aphandra natalia* and *Gustavia macarenensis* reported α-tocopherols concentrations of 21.52 ± 1.30 and 90.23 ± 2.32 mg/Kg, respectively [17,27]. These compounds increase when there is high light, salinity, nutrient deficiency, and cold, with α-tocopherol being the most potent form of vitamin E [97].

## 3. Discussion

### 3.1. Ethnobotany

Orellana, Sucumbíos, and Napo are the provinces of the Ecuadorian Amazon with the most records of useful species. In Ecuador, fruits are part of the plants mostly used by animals as food, and it has been recorded that the animals that mainly feed on fruits are birds, followed by mammals, fish, and reptiles [71]. Fruits are commonly viewed as a healthy food choice due to their natural sweetness, making them a popular snack throughout the day [98]. The local community recognizes fruits from the Amazonian biodiversity as a source of micronutrients that promote good health [99]; it has been observed that children who spend less time with their families in natural settings tend to prefer commercial fruit over wild fruit [100]. However, it is important to note that children may grow tired of the basic flavors they initially enjoy and may develop a taste for more intricate flavors [101,102]. Therefore, there is growing interest in “superfruits” that possess high nutraceutical value by containing bioactive compounds, which encourages the consumption of affordable foods from the local biome [103].

The second category of use of fruits is medicinal, followed by animal feed and materials. Regarding the use of fruits as medicine, younger generations may be less familiar with their properties, while women tend to be the main source of traditional knowledge and plant conservation. Communities located far from health centers tend to have more knowledge about plant use due to limited transportation and resources for accessing medicine. This contrasts with communities located closer to health centers [38].

In Ecuador, fruits, seeds, and leaves are the most consumed parts of the food species [71]. Interestingly, the peels of fruits are often discarded despite their potential pharmacological benefits. These peels contain bioactive compounds like polyphenols, flavonoids, carotenoids, terpenoids, and alkaloids, which have antioxidant, antiviral, anticancer, and antidiabetic properties [104]. They can be used for various purposes, such as raw materials for energy generation like bioethanol, biomethane, biohydrogen, and bio-oil. Additionally, bioactive chemicals can be extracted from them, and they can also be used as biofertilizers or bio-adsorbents [105].

### 3.2. Physicochemical Characterization

The fruits from the Ecuadorian Amazon that reported morphometric weight data (Table 3) are mostly round or oval, except for *Aphandra natalia* and *Plukenetia volubilis,* which are irregularly shaped. Morphometric characterization of fruits and seeds is essential for taxonomic identification and verification of environmental, phenotypic, and genetic variation in plant populations [106]. Fresh fruits are considered highly perishable food products because they have a high moisture content, and if not handled properly, there is accelerated deterioration [107]. Fruit weight is a quality trait that allows investigation of domestication and species improvement [108]. Pulp yield was reported for three Ecuadorian Amazonian fruits; this parameter can be related to the amount of total soluble solids, acids, salts, and other substances dissolved in the aqueous phase; therefore, fruits with higher values are suitable for processing [109].

Fruits from the Ecuadorian Amazon are consumed by local people, especially for their sweet and exotic flavor. For farmers, fruit trees are not the main crops; rather, they are species that are kept on the farm to diversify products for self-consumption and to market their surplus at local fairs during the harvest season. The Amazon fruit season in Ecuador occurs once a year between November and March. Indigenous women are the main traders of Amazonian fruits and are the ones who report uses in addition to food, such as cosmetic and medicinal use. The difficulties that exist for the use of these fruits are the long distances that must be traveled to harvest them, the fact that they are highly perishable products, the scarce and inadequate transportation, and the low, unregulated prices. Despite this, it is encouraging that the younger generations still like Amazonian fruits, and there is a high demand from local people during the harvest season.

In the nutritional composition of the fruits studied in the Ecuadorian Amazon (Table 3), fiber and lipid content are relevant for *Aphandra natalia*, *Gustavia macarenensis,* and *Plukenetia volubilis.* The fat content of *Aphandra natalia* sampled in the Amazon region of Ecuador (57.92 ± 3.05 dry wt%) is higher than that of fruits of the same Aracaceae family, such as *Mauritia flexuosa* (38.38 dry wt%) and *Oenocarpus bataua* (21.65 dry wt%) sampled in the Brazilian Amazon [110]; however, they coincide in the lipid profile reporting a high oleic acid content (*Aphandra natalia* 71.92%; *Mauritia flexuosa* 75.7%; *Oenocarpus bataua* 76.8% in dry wt), so this fruit can be considered a source of high-quality vegetable oil [17]. In contrast, *Gustavia macarenensis* has a lipid profile composed of palmitic (42.89%) and oleic (43.30%) acids [42].

The fat component of *Plukenetia volubilis* is characterized by a high content of polyunsaturated (51.5%) and monounsaturated (32.5%) fatty acids, which differ from a Peruvian accession that shows values of 46.6% and 36.5%, respectively. This is probably due to influential factors such as genetic differences and growing conditions like elevation and temperature [111]. The high carbohydrate content of *Eugenia stipitata* can be explained by the amount of total sugars (50% dry wt%) and dietary fiber (39 dry wt%) [112]. The protein content of over 10% in *Eugenia stipitate*, *Gustavia macarenensis*, *Plukenetia volubilis*, and *Solanum quitoense* is high compared to the 3.5% generally present in fruits [112]. Fruits such as *Myrciaria dubia* and *Pouteria caimito* report low lipid and protein values but have benefits such as reduced oxidative stress and plasma cholesterol, respectively [47,56]. Despite the nutritional content of Amazonian fruits, which can be considered underutilized fruits, their industrial and commercial application is limited because they are seasonal crops with a short shelf life [110].

### 3.3. Bioactive Compounds and Antioxidant Capacity

Antioxidant capacity is reported according to the method of analysis, which makes it difficult to make comparisons. However, it is a measure that allows establishing a reference on the capacity of a food to eliminate free radicals, which cause cell damage and are correlated with mutagenesis, carcinogenesis, and aging [113]. This identifies the importance of food consumption as part of a diet with protective effects for health and the need for research on the components responsible for this capacity, such as vitamin C, phenolic compounds, and carotenoids [114]. According to [115], the non-traditional tropical fruits from Brazil that stood out for their antioxidant capacity were puçápreto (EC50 = 414 and 65.6 g/g DPPH), camu-camu (EC50 = 478 and 42.6 g/g DPPH), and acerola (EC50 = 670 and 49.2 g/g DPPH), which correlated with the phenol content and specifically for acerola and camu-camu with the vitamin C content. Fruits selected for this review that report this parameter and highlight their high level of vitamin C were *Eugenia stipitata*, *Gustavia macarenensis*, *Oenocarpus bataua*, and *Pouteria caimito,* confirming the potential of underutilized tropical fruits as functional foods.

Fruits and vegetables are the best natural source of vitamin C for humans. Vitamin C is an antioxidant capable of terminating free radical chain reactions through disproportion to non-toxic and non-radical products [114]. The fruits reported in Table 4 contain low concentrations of vitamin C, except *Myrciaria dubia (*camu-camu*),* considering that most tropical fruits have concentrations higher than 1000 mg/100 g. For instance, there are poorly known fruits present in the Amazon biome that have the highest reported concentrations; among them are acerola (1600 mg/100 g) and camu-camu (1882 ± 43.2 mg/100 g) [115].

Phenolic compounds are the most abundant dietary antioxidants that protect mammalian systems. Tropical fruits are rich sources of polyphenols compared to temperate climate fruits, especially those with polyphenol concentrations greater than 1000 mg GAE/100 g [114]. Of the fruits described in this study, the content of Eugenia stipitata and *Myrciaria dubia* pulp, as well as the contents of the peel of *Gustavia macarenensis*, *Myrciaria dubia*, and *Oenocarpus bataua,* was higher than 1000 mg GAE/100 g. *Mauritia flexuosa* and *Solanum quitoense* have values above 700 mg GAE/100 g, which are higher than those reported for *Mauritia flexuosa* samples from two Brazilian localities (362.90 ± 7.98–435.08 ± 6.97) [116] and lower than the value for *Solanum quitoense* collected in the province of Pichincha, Ecuador (1008 mg of GAE/100 g d.w.) [68]. *Oenocarpus bataua* and *Gustavia macarenensis* showed values above 600 mg GAE/100 g for the pulp, although the concentration was higher in their peels; a similar result showed the pulp of *Oenocarpus bataua* collected in the city of Pucallpa in the Peruvian Amazon (672.3 ± 46.9). The variation in phenol concentrations of fruits from different localities may be due to climatic and agronomic factors such as light exposure, rainfall, relief, soil type, location, crop variety, harvest time, crop management, maturity stage of the product, type, and storage conditions of the harvest, etc. [117].

Flavonoids constitute the largest class of polyphenols. The fruits in which there was a higher number of flavonoids were *Solanum quitoense* and *Eugenia stipitata*, which report higher values than products considered rich in these nutrients, such as *Allium fistulosum* L (272.05 mg/100 g of dry weight), *Psidium guajava* L. (112. 85 mg/100 g), and Camellia chinensis Kuntze (149.10 mg/100 g) [118], but, they had lower concentrations than pomegranate peel (>2000 mg QE/100 g d.w.), which is characterized by its potential as a natural antioxidant [104]. For *Gustavia macarenensis* and *Oenocarpus bataua*, it is highlighted that the flavonoid content is higher in the peels than in the edible part of the fruit.

Carotenoids are very versatile bioactives because they are precursors of vitamin A, pigments, antioxidants, and health-promoting compounds. In fruits, carotenoids such as xanthophylls are found in greater quantities and are mainly housed in the chromoplasts of ripe fruits [119] and provide yellow to red color to plant tissues, thus attracting consumers [120]. The Ecuadorian Amazon *Eugenia stipitata* and *Solanum quitoense* were the fruits that reported higher contents of total carotenoids (31 and 34.8 µg β-carotene/g respectively); however, Amazonian fruits studied in Brazil, such as *Mauritia flexuosa* (234–713 µg/g) and Bactris gasipaes (26–357 µg/g) [20], showed higher concentrations. In addition, it is important to consider that the presence of lipids is relevant for the bioavailability of carotenoids and that the concentration of these compounds varies depending on the origin, environment, climate, soil [116], ripening stage, characteristics, and storage time [76].

Anthocyanins are flavonoid compounds found in plant tissues and are characterized by their antioxidant properties and pigmenting power. In fruits, they constitute one of the pigments responsible for the color change in the ripening process, which allows their identification for animal consumption and the distribution of viable ripe seeds [120,121]. Pigmentation by anthocyanins decreases with high temperatures and increases with cold temperatures and in conditions of high light intensity. In this review, the fruits that recorded these compounds were *Gustavia macarenensis*, *Mauritia flexuosa*, *Myrciaria dubia,* and *Oenocarpus bataua,* while *Solanum quitoense* and *Eugenia stipitata* indicate that these compounds were not detected. However, the reported ranks are low with respect to the anthocyanin content of blackberries (81.3 mg/10 g FW), black grapes (41 mg/10 g FW), and raspberry (22.8 mg/10 g FW), which are anthocyanin-rich [122].

Tocopherols can scavenge free radicals and protect plant tissues from lipid peroxidation and oxidative damage. The concentration of pulp oil from the fruits *Aphandra natalia* and *Gustavia macarenensis* recorded in Table 4 would be below values for olive fruits (267 ± 7.3 mg/kg d.w.) [123], in a similar range detected for mandarin (50–150 mg/kg) where α-tocopherols is the predominant form of total tocopherols (over 70%) as in grapefruit and orange [97], and higher than reported for passion fruit, considering that in this fruit α-tocopherols were not detected but β-γ-δ tocopherols were present (0.52–0.61 mg/kg). It is relevant to indicate that tocopherols presented a decrease in the passion fruit crop that was conventionally managed with respect to organic management [124]. In the olive crop, these compounds also decreased with the application of fertigation; however, during cold storage of mandarins, the concentration was maintained [123].

Considering the characteristics and properties of native Amazonian fruits, it is crucial to expand research on their nutraceutical functions and the domestication process of these species, which begins with the exploitation of wild plants, continues with the cultivation of plants selected by nature, and ends with the fixation of morphological and genetic characteristics through human selection. This will allow for the establishment of appropriate strategies for using genetic resources, as well as for the conservation and extraction of non-timber resources [125,126].

## *4.* Materials and Methods

### 4.1. Study Area

The study area for this research was the Ecuadorian Amazon region, which, according to the political–territorial division of the country, is subdivided into 6 provinces with administrative autonomy: Sucumbíos; Napo; Orellana; Pastaza; Morona Santiago; and Zamora Chinchipe [127]. The importance of this region in Ecuador is high because it constitutes 46% of the national territory (120,000 km) and has a population of 956,699 inhabitants, according to the projection for 2020 and considering data from the 2010 population census [128].

### 4.2. The Systematic Literature Review

The systematic review of the literature conducted from June to August 2023 was carried out following the guidelines for the Preferred Reporting Items for Systematic Reviews and Meta-Analyses (PRISMA) [129]. Briefly, the research question was formulated; then, the inclusion/exclusion criteria and the databases for the search were established to continue with the filtering, extraction, synthesis, and interpretation of the data obtained, and finally, the discussion of the results was performed.

The databases used were Scopus, Web of Science, PubMed, Scielo, and Redalyc. Articles published up to August 2023 were included. The keywords on which the search was based are described in Table 5. The inclusion criteria were articles that sampled fruits in the Ecuadorian Amazon region. These documents were used to identify the fruits studied in the Ecuadorian Amazon region (55 fruits), which were listed with their ethnobotanical characteristics (Appendix A). From this list of 55 fruits, fruits of indigenous origin were selected if they had at least two references from the first search of the review (*Aphandra natalia*, *Eugenia stipitate*, *Gustavia macarenensis*, *Mauritia flexuosa*, *Myrciaria dubia*, *Oenocarpus bataua*, *Plukenetia volubilis*, *Theobroma cacao*); however, *Theobroma cacao* was excluded because it is a traditional export product. The search for information on the 9 selected fruits was carried out using the keywords in Table 5, subject fruit description.

The search for the identification of edible fruits studied in the Ecuadorian Amazon region yielded a total of 310 documents, which were screened, including only 27 (Figure 2). In addition, 738 documents resulted from the search carried out to describe the physicochemical characteristics and bioactive components of the selected fruits. These documents were also screened by a title/abstract and full document reading, including only 39 documents in the review of the selected fruits: *Aphandra natalia*; *Eugenia stipitate*; *Gustavia macarenensis*; *Mauritia flexuosa*; *Myrciaria dubia*; *Oenocarpus bataua*; and *Plukenetia volubilis.* The inclusion criterion for the second search was the 10 most-cited documents.

## 5. Conclusions

The Ecuadorian Amazonian fruit trees with the most studies available were *Aphandra natalia*, *Solanum quitoense*, and *Eugenia stipitata*. The analysis of the ethnobotanical information identified that the most used part of the fruits is the pulp, that the most frequent uses of the fruits are fresh consumption, preparation of typical dishes, and medicinal use, and that Sucumbíos and Orellana are the Amazonian provinces where the fruits have been investigated the most.

Within the physicochemical characteristics of fruits, morphometric and nutritional content information is the most regularly reported. The fiber, fat, and protein contents of the Amazonian fruits *Aphandra natalia*, *Solanum quitoense*, *Gustavia macarenensis*, and *Plukenetia volubilis* were relevant with respect to what the fruits generally present, which demonstrates their potential as an alternative for the food industry through their processing, considering their high susceptibility and seasonal production.

The bioactive components analyzed in all fruits were total polyphenols. *Solanum quitoense* and *Eugenia stipitata* were the fruits with the highest content of total flavonoids and carotenoids in the pulp, while *Gustavia macarenensis* and *Oenocarpus bataua* reported higher amounts of anthocyanins in the pulp and flavonoids in the peel. The amount of these bioactive components reported in Ecuadorian Amazonian fruits are like products characterized by their high content, which allows asserting the nutraceutical capacity of these products.

Considering the 55 edible fruits identified and 9 fruits selected for this bibliographic review, there is a great need to expand research on quality parameters, bioactive components, and domestication of native Amazonian fruits to establish nutritional alternatives with nutraceutical properties and the commercialization of non-timber resources that can contribute to the sustainable development of the region.

## Figures and Tables

**Figure 1 plants-12-03635-f001:**
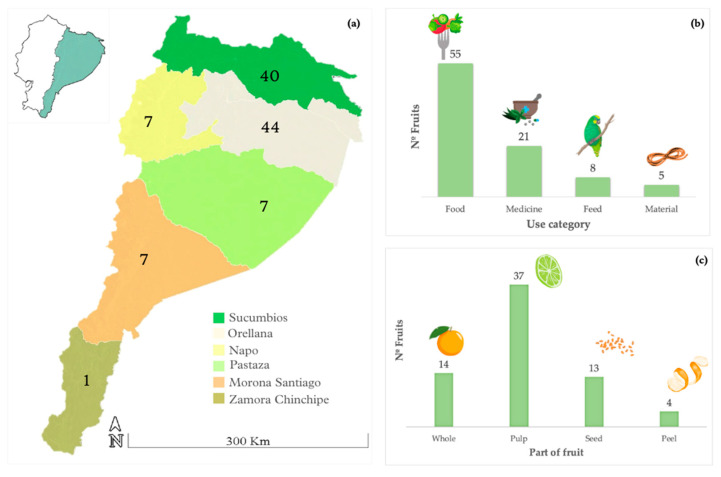
(**a**) Location of number of fruits reported by province in the Ecuadorian Amazon. (**b**) Fruits use categories identified. (**c**) Part of the fruit used and reported in the investigations.

**Figure 2 plants-12-03635-f002:**
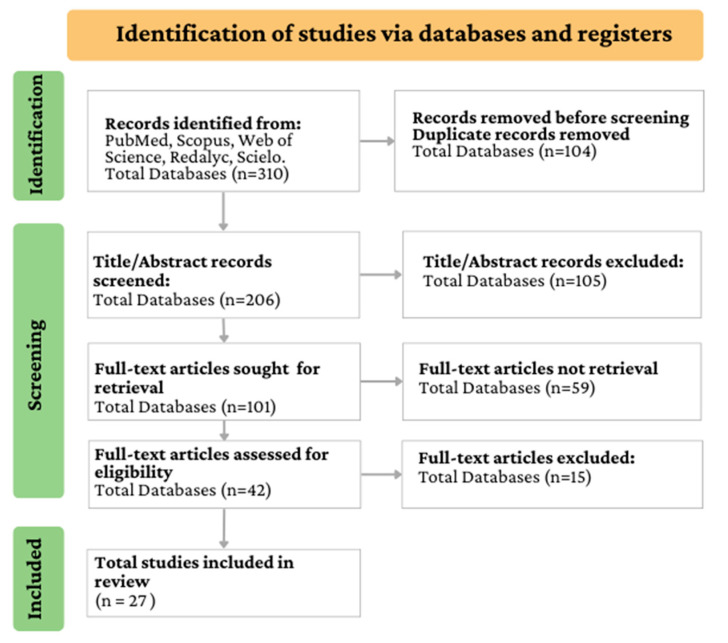
Schematic diagram of identification, screening, and included dataset through PRISMA.

**Table 1 plants-12-03635-t001:** References included by topic.

Fruits	References Included
*Aphandra natalia*	[17]
*Eugenia stipitata*	[32,43,44,45,46,47,48,49,50]
*Gustavia macarenensis*	[27,42,51]
*Mauritia flexuosa*	[21,25,41,44,47,52,53]
*Myrciaria dubia*	[46,54,55,56,57,58]
*Oenocarpus bataua*	[26,27,59,60]
*Plukenetia volubilis*	[61,62,63,64,65]
*Pouteria caimito*	[47,66,67]
*Solanum quitoense*	[32,33,43,68,69]

**Table 2 plants-12-03635-t002:** Description of fruit species of the Ecuadorian Amazon with information on physicochemical characteristics.

Fruit	Description
** * 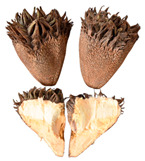 * **	*Aphandra natalia* (Balslev and Henderson) Barfod is a species of palm native to the Ecuadorian Amazon; it is characterized by having a hard, scaly rind, slightly sweet, fibrous pulp with high-fat content, and large seeds that are used for making handicrafts. The mesocarp oil of this fruit has a high oleic acid content (71.92%). However, in Peru, Brazil, and Ecuador, the extraction of fibers for the manufacture of brooms and the collection of leaves for roofing are the most frequent uses for this species [70]. It grows on terraces along rivers at an altitude of up to 800 m but can be cultivated at up to 1000 m above sea level. This species is found in a wide range, extending from south of the Napo River in Ecuador to the lowlands of the western Amazon basin, across the northern Peruvian Amazon to the Acre in Brazil. This species can also be found in home gardens and agroforestry systems, where it provides shade for livestock and protects the soil from erosion [71], but its low densities are due to its dispersal rather than environmental limitations [72]. Family: Arecaceae|Common name: Tagua; Vegetable ivory.
** * 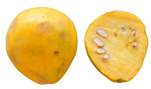 * **	*Eugenia stipitata* McVaugh has an oval-shaped climacteric berry fruit with a sweet and sour taste and a very pleasant characteristic aroma, which turns from green to yellow or orange when ripe, with three harvests per year. The fruits are used for the preparation of homemade juices, although they also have medicinal uses [43]. This species is native to western Amazonia and the Guianas; it grows in tropical and subtropical climates and is cultivated in Bolivia, Brazil, Colombia, Ecuador, and Peru. It is an approximately 3-meter shrub with dense foliage that grows in fertile, well-drained soils. [49]. Family: Myrtaceae|Common name: Araza; Amazonia guava.
** * 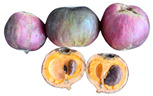 * **	*Gustavia macarenensis* Philipson is a type of fruit that has a soft, orange pulp with a high-fat content. It is known for its pleasant taste when eaten fresh and can also be used as a purgative. The fruit’s shell can be infused and used as a uterine antihemorrhagic, while the fresh seeds can help counteract dysentery and sinusitis. This makes it a promising fruit for applications in the food, cosmetic, and pharmaceutical industries [27]. This species can be found at various altitudes, ranging from 260 to 1200 m above sea level. Its habitat in Ecuador covers all provinces of the Amazon region. The tree can grow up to 25 m tall, with a dense and rounded crown. Its fruits are ripe when they fall off the tree [42]. Family: Lecythidaceae|Common name: Alan paso; Inaco; Inak; Passo; Sachu pasu.
** 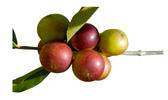 **	*Myrciaria dubia* L.f. berry is round and contains 1 to 4 seeds, with a diameter of approximately 2.5 cm. The color of the berry changes from green to violet–red as it ripens, which is due to the presence of phenolic compounds [54]. This fruit contains various bioactive components like anthocyanins, flavonols, ellagitannins, proanthocyanins, and carotenoids [58]. Peru mainly exports this fruit in the form of flour, extract, and dehydrated products due to its economic potential. The species is native to the Amazon rainforest in Brazil and grows naturally on the banks of rivers and lakes during periods of flooding [73]. Family: Myrtaceae|Common name: Camu-camu; Guayabo; Guayabito; Guapuro blanco.
** * 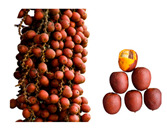 * **	*Mauritia flexuosa* (Kunth) McVaugh is a palm that constitutes a socioeconomic resource for local indigenous communities in the Ecuadorian Amazon, growing in poorly drained soils and generating resources such as food, nesting sites, and habitat [74]. This fruit is a highly traded item in local markets. Its vibrant orange pulp is commonly used to make sweets, ice cream, juices, jams, and oil. It is also a rich source of bioactive compounds, with a higher β-carotene content than such products as carrots and spinach. As a result, it can be classified as a functional food [21]. This species has a broad distribution in South America, primarily in the Amazon region, spanning across Bolivia, Brazil, Colombia, Ecuador, Guyana, Peru, Trinidad and Tobago, and Venezuela [75]. It can be in various habitats, ranging from lowland tropical forests to open savannah landscapes, floodplains, and springs [76]. The tree is a palm that grows up to 40 m tall, featuring a cylindrical and smooth stem with a diameter ranging from 30 to 60 cm. It has a spherical crown of elongated leaves reaching up to 6 m in length [77] Family: Arecaceae|Common name: Morete; Aguaje; Buriti; Moriche.
** * 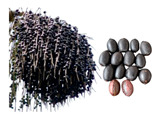 * **	*Oenocarpus bataua* Mart. tree produces fruit with a valuable violet-colored pulp. This pulp is rich in monounsaturated fats and antioxidants that can be extracted for use. Additionally, the fruit is used to make both fermented and non-fermented beverages and desserts [78]. They constitute a key element in the food security of Amazonian peoples and a promising element for the food and cosmetic industry [79]. This is a type of palm that thrives in tropical rainforests and is commonly found in the Amazon biome, including both flooded and mainland forests [60]. It can reach heights between 4 and 26 m and can yield up to eleven tons of fruit per hectare annually [27]. It is one of the most used palms in Amazonia, growing preferably in poorly drained soils; additional uses of it include straw, fiber, and wood [26]. Family: Arecaceae|Common name: Chapil; Patawa; Seje; Ungurahua.
** 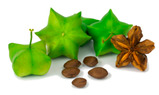 **	*Plukenetia volubilis* L. produces fruits whose seeds are appreciated for their high content of unsaturated fatty acids (omegas 3 and 6) and proteins (27–33%), which are rich in essential amino acids and consumed as nuts [80]. The leaves have antioxidant components such as flavonoids, terpenoids, and saponins [81]. This species is a shrubby climbing plant that originates from the Amazon rainforests and can thrive in warm climates, from the high altitudes of the Andean jungle to the lowlands of the Peruvian Amazon [82]. Its seeds contain essential amino acids, making it a promising economic crop in Central and South America, as well as in Southeast Asian countries [80]. Family: Euphorbiaceae|Common name: Sacha inchi; Sacha yuchi; Sacha yuchiqui; Mountain peanuts; Wild peanuts; Inca peanuts; Sacha peanuts.
** 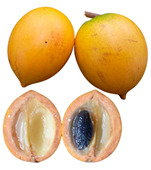 **	*Pouteria caimito* (Ruiz and Pav.) Radlk is a round berry that often has a pointed end with a smooth peel that contains latex. When it ripens, it changes from green to bright yellow and is a climacteric fruit that contains beneficial bioactive compounds [83]. However, it has a short shelf life due to browning and high levels of ethylene production [84]. The fruit’s pulp is translucent, white, juicy, soft, sweet, and mucilaginous. It is usually eaten raw, but it can also be used to make soft drinks, ice creams, sorbets, and other desserts. This tree is a perennial that can grow up to 5–15 m tall and is native to the Amazon region of South America [85]. It is also found in many other places, including Central America, the West Indies, Northern Australia, and Malaysia. It is grown on farms, in orchards and gardens, and in urban forestry in some regions of northern Brazil [86]. Family: Sapotaceae|Common name: Caimito; Cauje; Luma.
** * 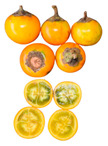 * **	*Solanum quitoense* Lam. is a perennial herbaceous plant native to Ecuador and Colombia; the fruit is edible and resembles a small orange with pubescence, sweet acid flavor, and exotic and very pleasant characteristic aroma [87]. In Ecuador, the crop is grown mainly in the Amazon region at the altitude of between 1500 and 2400 m; the common name is naranjilla; it is consumed in juice, although it is also marketed as pulp and has been used to produce bottled juices and ice cream [88]. This plant, a perennial shrub that can grow up to 2.5 m tall, is originally from the Andes and can be found from Venezuela to Peru [89]. It is typically grown at elevations between 1000 and 1900 m. There are two types of this plant: varietas quitoense, which does not have thorns and is found in Colombia and Ecuador, and varietas septentrionale, which has thorns and can be found in Colombia, Panama, and Costa Rica [68]. In Ecuador, it is primarily grown in the mountainous foothills and Amazonian plains for both local consumption and export to Colombia [90]. Family: Solanaceae|Common name: Naranjilla; Lulo.

**Table 3 plants-12-03635-t003:** Physicochemical characteristics of edible fruits native to Ecuadorian Amazonia.

Fruit	Length (cm)	Transverse Diameter (cm)	Weight (g)	Moisture (%)	Pulp Yield (%)	Total Soluble Solids (°Brix)	Titrable Acidity (%)	Maturity Index	Ash (%)	Crude Fibre (%)	Lipids (%)	Proteins (%)	Carbohydrates (%)	Reference
*Aphandra natalia*	13.56 ± 4.97	Not applicable	522.77 ± 48.01	44.52 ± 0.11 (Mesocarp)	NR	NR	0.04 *	NR	2.62 ± 0.05 *	11.30 ± 0.59 *	57.92 ± 3.05 *	5.11 ± 0.10 *	23.0 ± 2.35 *	[17]
*Eugenia stipitata*	5.54	7–12	30–105	80.30–94.42	89.65 (pulp + peel)	3.83–5.06	2.40 ± 0.02	1.60 ± 0.45	2.1–2.94 *	6.69–8.24 *	0.59–3.02 *	9.92–12.68 *	60.36–64.54 *	[32,43,44,45,46,47,48]
*Gustavia macarenensis*	5.73–7.04	6.09–8.50	153.93–280.18	61.74–70.94	59.34	60.05–69.8 (oil pulp)	NR	NR	2.34–4.08	11.53–25.17	31.60–53.57	12.39–12.8	19.73–26.75	[42,51]
*Mauritia flexuosa*	5.47 ± 0.15	4.59 ± 0.18	48.7–51.83	54.8–56.23 (pulp)	11.0–20.19	7.73 ± 0.06	7.60 ± 0.23	1.07	2.27 ± 0.05	38.0 ± 0.3	26.6 ± 0.3	2.47 ± 0.07	15.1 ± 0.3	[21,25]
*Myrciaria dubia*	1.2–2.5	1.0–3.2	6.9	93.2	50–60	6.2–9.39	2.30–4.97	2.14	1.8–2.64	0.1–1.3	0.2–0.3	0.4–0.5	3.5–4.7	[46,54,55,56]
*Oenocarpus bataua*	33.3–35.9	23.9–24	13.23	42.4	11.84	NR	NR	NR	1.1	31.5	12.8–22.2 38.3 *	3.3	47.2	[26,59,60]
*Plukenetia volubilis*	3–5 (star-shaped capsule)	1.5–2 (oval seed)	NR	3.3–8.32 (seed)	NR	NR	NR	NR	2.9–6.46	18.0 ± 0.095	33.4–54.3	24.2–29.7	7.29–30.9	[61,62,63]
*Pouteria caimito*	NR	NR	NR	81.87–95.8	NR	3.8–17.50	0.04–5.9	NR	0.32–0.49	NR	0.1–0.15	4.60–4.97	NR	[47,66,67]
*Solanum quitoense*	4.7–5.24	5.4–5.89	65.7–311.54	90.74 ± 0.73 (whole fruit)	63.01–84.72	6.53–9.55	1.85–3.78	1.83–3.72	5.84 ± 0.85	9.22 ± 0.22	0.69 ± 0.05 11.65 *	11.2 ± 1.99 7.44 *	NR	[32,33,68]

* Values expressed in dry weight (values without * are expressed in fresh weight); NR: Not reported.

**Table 4 plants-12-03635-t004:** Bioactive compounds content of edible fruits of the Ecuadorian Amazon.

Fruit (Scientific Name)	Antioxidant Capacity	Vitamin C (mg/100 g)	Total Phenolics (mg GAE/100 g)	Total Flavonoids (mg/100 g)	Total Carotenoids (mg β-Carotene/kg)	Total Anthocyanin (mg/100 g)	Reference
		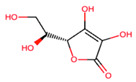	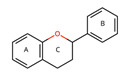	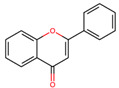	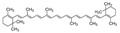	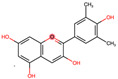	
*Aphandra natalia*	DPPH: 60 ± 1.0 µmol TE/kg (pulp oil)	NR	8.36 ± 1.84 (pulp oil)	NR	0.24 ± 0.02	NR	[17]
*Eugenia stipitata*	DPPH: 392.10–3621.8 * µmol TE/kg (pulp) ABTS: 758.22 ± 5.01 * µmoL TE/g (pulp) DPPH: 84.4 μmol TE/100 (pulp) FRAP: 63 μmol FeSO4/g (pulp)	5.60–8.3	14,243 (seed) 154.20 (pulp)	125–600.72 * (pulp) 4373 ± 0.23 (seed)	31.00–33.39 (pulp)	ND	[32,43,44,46,47,49,50]
*Gustavia macarenensis*	DPPH: 688.40 ± 2.28 * µmol AA/100 g (pulp) DPPH: 1146.41 ± 1.12 * µmol AA/100 g (peel) DPPH: 523.82 ± 14.09 * µmol AA/100 g (seed) DPPH: 170 ± 2.86 μmol TE/Kg (pulp oil)	5.48 ± 0.24 * (pulp) 15.85 ± 0.06 * (peel) 2.20 ± 0.11 * (seed)	634.30 ± 5.01 * (pulp) 1165.87 ± 17.66 * (peel) 153.16 ± 3.45 * (seed) 156.49 ± 2.62 (pulp oil)	25.57 ± 0.60 * (pulp) 383.59 ± 8.136 * (peel) 111.39 ± 0.79 * (seed)	2.62 ± 0.16 (pulp)	25.57 ± 0.60 * (pulp) 9.13 ± 0.11 * (peel) 5.79 ± 0.12 * (seed)	[27,42]
*Mauritia flexuosa*	DPPH: 14.7–17.98 * µmoL TE/g (pulp) FRAP 11.38–15.68 * µmoL TE/g (Pulp)	3.28 ± 0.78 33.4 ± 0.48 * (pulp) 59.93	725.0–743.2 * (pulp) 29.026–174.36 (shell)	55.8–196 * (pulp) 20.27–185.58 (shell)	239.4–216.4 (pulp)	3.10 mg cyanidin 3-O-glucoside 100 g^−1^	[25,41,52,93]
*Myrciaria dubia*	DPPH: 1.93 μmol TE/g (ripe fresh pulp) DPPH: 1328.50 ± 14.40 μmol TE/g (ripe peel)	960–4752.23 (pulp) 1109.62 ± 56.5 (ripe peel)	12,798.80 (pulp) 13,348.97 ± 99.94 (ripe peel)	55.1–211.64 (pulp)	1 (pulp) 6199.8 (ripe peel)	170.00 (pulp) 223.14 ± 6.89 (ripe peel)	[57,58,94]
*Oenocarpus bataua*	DPPH: 478.94 ± 4.85 * µmol AA/100 g (pulp) DPPH: 654.56 ± 0.94 * µmol AA/100 g (peel) DPPH: 589.44 ± 5.29 * µmol AA/100 g (seed)	0.12 ± 0.00 (pulp)	622.97 ± 4.84 * (pulp) 1009.38 ± 1.80 * (peel) 758.25 ± 3.221 * (seed)	55.34 ± 0.29 * (pulp) 57.17 ± 0.42 * (peel) 47.15 ± 0.06 * (seed)	NR	14.82 ± 0.20 * (pulp) 46.48 ± 0.31 * (peel) 14.71 ± 0.02 * (seed)	[27]
*Plukenetia volubilis (seed)*	DPPH: 32.43 ± 1.63% (Oil pressed-cake) FRAP: 732.67 ± 35.29 µmol FeSO_4_/L (Oil pressed-cake) ORAC: 6.5–9.8 μmol TE/g	NR	6.58 ± 0.27 (oil seed) 51 (Oil pressed-cake)	NR	NR	NR	[65]
*Pouteria caimito*	DPPH: 734.98 ± 0.26 µmol TE/100 (pulp) FRAP: 1211.03 ± 1.12 µmol TE/100 (pulp)	2.0–3.11 (pulp)	172.75 ± 0.77 (pulp) 60.0 ± 0.015 (pulp)	NR	0.25 (pulp)	NR	[47,66,67]
*Solanum quitoense*	DPPH: 21.26 ± 1.35 * µmol TE/g (pulp) ABTS 76.40 ± 1.33 * µmol TE/g (pulp) ORAC: 0.15 ± 0.02 mM/g (Pulp)	0.1 * (whole fruit) 4.16 ± 1.49 (pulp)	775.31–897.58 *	991.57 ± 20.24	34.08 ± 2.42–57.93 *	ND	[32,43,69]

* Values expressed in dry weight (values without * are expressed in fresh weight). Values expressed as ng/g of dry weight; GAE, gallic acid equivalent; TE, trolox equivalent; AA, ascorbic acid equivalent; ND, not detectable; NR, not reported.

**Table 5 plants-12-03635-t005:** Search parameters of the literature review.

Subject	Search Parameters
Fruits	“fruit” AND “Amazon” AND “Ecuador”
Ethnobotany	“ethnobotany” AND “Ecuador” AND “Amazon”
Bioactive compounds	“bioactive” AND “Ecuador” AND “Amazon”
Fruit description	“scientific name of fruit” AND “physicochemical” “scientific name of fruit” AND “bioactive”

## Data Availability

No new data were created or analyzed in this study. Data sharing is not applicable to this article.

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
