# Peer review of "Edible Fruits from the Ecuadorian Amazon: Ethnobotany, Physicochemical Characteristics, and Bioactive Components"

_plants, 2023, doi:10.3390/plants12203635_

Round 1
Reviewer 1 Report
This work and the subject it deals with are interesting, but I think it should be better structured and more carefully written if it is to be accepted as it stands.
With regard to the literature search and the selection of articles for the revision, I think it would be advantageous to use a standardized methodology that has already been described (e.g. PRISMA http://www.prisma-statement.org/?AspxAutoDetectCookieSupport=1).
The search for information on nutritional properties seems to have been carried out only in publications from Ecuador. As these species can also occur in neighboring countries, it would be advisable to check whether other edible wild species recorded in Ecuador have been studied in other South American countries.
Regarding the authors of the species, they should only be mentioned once, either in the first citation of the species or in a table, for example.
The common names of the species would also be welcome, for example in a table with the main characteristics of the species, including ecology and habitat.
Children are often the ones who consume the most wild fruits, so I think this issue should be better addressed.
A short discussion on the potential for domestication or co-domestication of some species would also be welcome, especially since one of the species mentioned has already been effectively domesticated.
Several other comments and suggestions can be found in the manuscript's doc.

The writing in English seems to be OK, but there was a lack of care in the translation (there are words and phrases in Spanish) and in the scientific names of the species (italics and names of authors of species).
Author Response
Dear reviewer, we appreciate the contributions made, this allowed us to significantly improve the manuscript. It is important to mention that the order of the sections of the manuscript was changed according to the request of the journal as follows:
- Introduction
- Results
- Discussion
- Materials and Methods
- Conclusion
- Patents
Comment 1. With regard to the literature search and the selection of articles for the revision, I think it would be advantageous to use a standardized methodology that has already been described (e.g. PRISMA http://www.prisma-statement.org/?AspxAutoDetectCookieSupport=1).
Reply 1. The methodology of the review followed the suggested PRISMA methodology. This can be found in lines 342 to 382.
Comment 2. The search for information on nutritional properties seems to have been carried out only in publications from Ecuador. As these species can also occur in neighboring countries, it would be advisable to check whether other edible wild species recorded in Ecuador have been studied in other South American countries.
Reply 2. The search had been carried out in documents whose methodology indicated that the fruits had been collected in Ecuador. Because of your suggestion, the initial search was used for the identification of the fruits studied in Ecuador from which 9 were selected and the search parameters for the physicochemical characteristics and bioactive components of the fruits were expanded to: scientific name of the fruit and the words "physicochemical" and "bioactive" which are the focus of the paper. This is described in the methodology. Lines 362-379
Comment 3. Regarding the authors of the species, they should only be mentioned once, either in the first citation of the species or in a table, for example.
Reply 3. We proceeded as indicated and the authors of the species are only mentioned the first time the name appears.
Comment 4. The common names of the species would also be welcome, for example in a table with the main characteristics of the species, including ecology and habitat.
Reply 4. Common names as well as ecology and habitat details of each species were added for each species in Table 2. The structure of the description of each fruit was established first details of the fruits, then ecology and habitat data, and finally common names.
Comment 5. Children are often the ones who consume the most wild fruits, so I think this issue should be better addressed.
Reply 5. Suggestion increased. Children's interest in wild fruits was highlighted in the discussion section. Lines 175-181
Comment 6. A short discussion on the potential for domestication or co-domestication of some species would also be welcome, especially since one of the species mentioned has already been effectively domesticated.
Reply 6. The suggested discussion was augmented on lines 334-340.
Comment 7. Several other comments and suggestions can be found in the manuscript's doc.
Reply 8. All suggestions described in the document were fully implemented.
Reviewer 2 Report
Dear Authors,
I have a few comments:
1. Latin names should be standardized: either all in italics or without italics, depending on the requirements of the journal;
2. There are also a lot of mistakes with use of dash, eg. line 98 ana-lyzed; line 139 should be: well-being; line 193 be-cause; line 201 fami-ly; line 247 compo-nents; line 301 Gus-tavia; line 305 pig-ments; line 308 Eugen-ia; line 310 Ama-zonian; line 312 concen-tration; line 329 re-ported; line 332 cardiopro-tective; line 335 consid-ered; line 339 nu-trient; line 341 Philip-son; line 342 respec-tively; line 344 predom-inant; line 349 or-ganic, de-creased; line 356 consump-tion; line 360 con-tent; line 361 qui-toense.
3. line 352: should be Conclusions
4. page 6, line 170 the sentence should be in English
5. I think, it is a good idea to show some chemical structures of the most popular natural compounds, that we can find in described fruits
Language generally is good
Author Response
Reviewer 2
Dear reviewer, we appreciate the contributions made, this allowed us to significantly improve the manuscript. It is important to mention that the order of the sections of the manuscript was changed according to the request of the journal as follows:
- Introduction
- Results
- Discussion
- Materials and Methods
- Conclusion
- Patents
Comment 1. Latin names should be standardized: either all in italics or without italics, depending on the requirements of the journal
Reply 1. Latin names were written all in italics following the suggestion.
Comment 2. There are also a lot of mistakes with use of dash, eg. line 98 ana-lyzed; line 139 should be: well-being; line 193 be-cause; line 201 fami-ly; line 247 compo-nents; line 301 Gus-tavia; line 305 pig-ments; line 308 Eugen-ia; line 310 Ama-zonian; line 312 concen-tration; line 329 re-ported; line 332 cardiopro-tective; line 335 consid-ered; line 339 nu-trient; line 341 Philip-son; line 342 respec-tively; line 344 predom-inant; line 349 or-ganic, de-creased; line 356 consump-tion; line 360 con-tent; line 361 qui-toense.
Reply 2. Se corrigieron todos los errores de guión.
Comment 3. line 352: should be Conclusions
Reply 3. Line 352 was corrected
Comment 4. page 6, line 170 the sentence should be in English
Reply 4. Line 170 was corrected. The modification is on lines 109-113.
Comment 5. I think, it is a good idea to show some chemical structures of the most popular natural compounds, that we can find in described fruits
Reply 5. The chemical structures were included in table 4.
Round 2
Reviewer 1 Report
The work has been greatly improved from the initial version and is now in a better position to be accepted.
I think it still needs some adjustments and corrections, namely:
- at least one reference (number 42) doesn't seem to be cited in the text, confirm this and the others
- fewer than 30 works were selected for the bibliographical review but around 160 are cited in total; check whether it makes sense to cite all these works
- some work is needed to standardise the references
- two species (Bactris and Carica) have been removed, probably because they are cultivated; if they can only be found in cultivation, I agree, but if they are naturalised or can be found in the wild, I think they could be included.
Please see some more comments in the PDF attached.

The English seems to be almost OK but can benefit from a careful revision (e.g. Referencias instead of References).
Author Response
Manuscript ID: Plants-2634615
Thank you very much for taking the time to review this manuscript, your contributions have been very helpful.
|
Comment |
Action |
|
Are the results clearly presented? The reviewer said: Can be improved
|
The suggestions made in this section corresponded to slight modifications of Table 2, which were executed and are in blue tail font. |
|
Are all the cited references relevant to the research? The reviewer said: Can be improved |
According to the indications both the relevance and the format of the references were corrected. When checking each of the citations, 30 were eliminated because some references had 2 or more different citations. In the references, the correct use of capital letters and the correct writing of scientific names were verified. |
|
At least one reference (number 42) doesn't seem to be cited in the text, confirm this and the others |
All references were verified, however, there are some like the one you mention (now reference 34, line 515) that appear because they are within the citation of the 27 documents used in the first search (line 77), which identified the 55 fruits studied in Ecuador (supplementary material). |
|
Fewer than 30 works were selected for the bibliographical review but around 160 are cited in total; check whether it makes sense to cite all these works |
The documents included in the review were 27 for identifying fruits studied in Ecuador and 39 for describing the 9 fruits selected as described in the methodology (line 371-377). In addition, we proceeded to verify that each of the citations in the document is linked to the corresponding reference. |
|
Some work is needed to standardise the references
|
The format of each of the references was verified. Considering that adjustments were made to the formatting of all references, the blue color of the font is evident. |
|
Two species (Bactris and Carica) have been removed, probably because they are cultivated; if they can only be found in cultivation, I agree, but if they are naturalised or can be found in the wild, I think they could be included. |
Considering the initial observations, the methodology was adjusted, in such a way that a first search was carried out that yielded 55 edible Amazonian fruits studied in Ecuador, then 9 native fruits were selected (according to https://bibdigital.rjb.csic.es/records/item/16016-enciclopedia-de-las-plantas-utiles-del-ecuador as specified in the supplementary material S1), and that in turn, had at least 2 references in the first review. In this selection both Bactris and Carica were discarded because they were not considered native to Ecuador. |
|
Please see some more comments in the PDF attached. |
The comments in the document were fully addressed. The name of the Family of the species was added at the end of the description of each fruit, in Table 2. |
